# Spatial Analysis of Tuberculosis Patient Flow in a Neglected Region of Northern Brazil

**DOI:** 10.3390/tropicalmed8080397

**Published:** 2023-08-02

**Authors:** Cristal Ribeiro Mesquita, Marília Lima da Conceição, Rafael Aleixo Coelho de Oliveira, Emilyn Costa Conceição, Juliana Conceição Dias Garcez, Ianny Ferreira Raiol Sousa, Luana Nepomuceno Gondim Costa Lima, Karla Valéria Batista Lima, Ricardo José de Paula Souza e Guimarães

**Affiliations:** 1Program in Parasitic Biology in the Amazon Region, State University of Pará, Tv. Perebebuí, 2623-Marco, Belém 66087-662, Pará, Brazil; juliana.garcez@famaz.edu.br (J.C.D.G.); raiolianny@hotmail.com (I.F.R.S.); luanalima@iec.gov.br (L.N.G.C.L.); karlalima@iec.gov.br (K.V.B.L.); 2Instituto Evandro Chagas, Bacteriology and Mycology Section, Ananindeua 67030-000, Pará, Brazil; marilimadc@msn.com; 3Graduate Program in Health/Environment and Society, Federal University of Pará, Belém 66075-110, Pará, Brazil; aleixorafaelll@gmail.com; 4DSI-NRF Centre of Excellence for Biomedical Tuberculosis Research, SAMRC Centre for Tuberculosis Research, Division of Molecular Biology and Human Genetics, Faculty of Medicine and Health Sciences, Stellenbosch University, Cape Town 7505, South Africa; emilyncosta@gmail.com; 5Instituto Evandro Chagas, Program in Epidemiology and Health Surveillance, Ananindeua 67000-000, Pará, Brazil; ricardojpsg@gmail.com

**Keywords:** epidemiology, tuberculosis, *Mycobacterium tuberculosis*, spatial analysis, social networks

## Abstract

Tuberculosis (TB) is still considered a priority due to its high incidence rate in Brazil. In this context, we aimed to evaluate the flow of care between the municipalities of patients diagnosed with TB through notification forms of the Information System for Diseases and Notifications (SINAN) in a neglected region of Northern Brazil, Ilha do Marajó, state of Para. For this, we performed a descriptive, retrospective study on data obtained from the National Register of Health Establishments and SINAN from 2013 to 2018. We used Pearson’s Chi-square and G Test with *p*-value < 0.05 for descriptive statistics and spatial analysis technique on flow network analysis. Of the 749 cases, 16.5% were notified in another municipality that was not the patient’s residence. Regarding diagnostic methods, a positive bacterioscopy was adopted for 56% of the patients; culture was not performed for 82% of cases. Histopathological examination was not performed in 90% of the individuals. Rapid molecular test (RMT) was performed in only six (5%) cases. The region needs greater attention focused on diagnostic tests, suggesting that the introduction of RMT and culture by Ogawa-Kudoh could improve the region’s health network to minimise patient displacement and thus avoid the increase in the transmission chain of TB.

## 1. Introduction

Tuberculosis (TB), caused by the *Mycobacterium tuberculosis* complex, remains a major cause of morbidity and mortality worldwide. Brazil is among the 30 countries with the highest TB burden globally [1]. Examining the historical data series of TB incidence in Brazil has revealed that, for at least the past decade, the state of Pará has consistently exhibited some of the highest incidence rates, surpassing the national average [2].

The National TB Control Programme (“*Programa Nacional de Controle da Tuberculose*”—PNCT) was initiated to control TB in Brazil, including the creation of Primary Health Care (PHC) to improve access to diagnosis, treatment, prevention, and surveillance for all users of the Brazilian Unified System of Health (“*Sistema Unico de Saúde*”—SUS). In this context, PHC has a fundamental role in the programme, as the Health Care Network (HCN) is the gateway for patients. However, health services are still conducted in a fragmented and episodic manner [3], mainly in neglected places regarding social policies.

The PNCT was established with the goal of disease control, implementing strategies to enhance diagnosis and detect symptoms at an early stage. This approach aims to disrupt the disease transmission chain, minimising the risks of illness and death and gradually achieving key indicators such as disease cure rates, reduced abandonment, mortality, comorbidity, and decreased time between symptom onset and diagnosis [4].

Individuals belonging to socially vulnerable groups are predominantly impacted by TB; thus, it is crucial to prioritise specific geographic areas to ensure equitable access to healthcare services. The proximity of healthcare facilities to patients’ residences plays a significant role in facilitating the early detection of respiratory symptoms. Furthermore, an active case-finding approach, along with meticulous attention to the performance of primary healthcare services and the effectiveness of health teams in investigating suspected TB cases, is essential for disease control. However, it is important to note that this patient care pathway may inadvertently contribute to increased disease transmission [5].

Sputum smear microscopy (bacterioscopy) and rapid molecular tests (RMT) are the main methods employed in detecting cases and early diagnosis of TB [5]. However, in addition to the difficulty in treating respiratory symptoms in PHC in Brazil, in specific locations, barriers to access and failures in care are of concern in populations living in regions lacking specialised services [5]. Ilha do Marajó (Marajó Island) in the state of Pará, a part of the Brazilian Amazon region, is a region of more than 40 thousand km^2^, and holds some of the municipalities with the worst Human Development Indexes (HDI) in Brazil [6].

Therefore, considering the need to enhance the healthcare network (HCN) in this region, this study aimed to comprehensively describe the epidemiological profile of TB cases in Ilha Marajó. For this, we analysed the patients’ municipality of residence and the municipality where TB cases were reported. The primary focus was evaluating the flow of attention to TB cases in Ilha do Marajó using data obtained from the Brazilian Notifiable Diseases Information System (SINAN) in the state of Pará.

## 2. Materials and Methods

### 2.1. Study Type

We conducted a descriptive, retrospective study that proposed to analyse the flow of TB patients in healthcare units to analyse and describe the inter-organisational structure characteristics of the health system and the networks of social interactions related to patients from TB in Ilha do Marajó.

### 2.2. Population and Study Site

The study population consisted of all new TB cases notified to the SINAN from 2013 to 2018. The study was conducted at the Ilha do Marajó, the largest island in this archipelago; it has 104,139.93 km^2^ in extension, is bathed to the northwest by the mouth of the Amazon River and is separated from the mainland to the south by the Pará River, which, to the southeast, expands receiving the waters of the Tocantins River and other smaller rivers, then changing its name to Baía do Marajó [7]. Its total population is 557,231 inhabitants, with a geographic density of 5.35 inhab/km^2^ [8].

### 2.3. Inclusion and Exclusion Criteria

The study encompassed all newly reported TB cases occurring in Ilha do Marajó between 2013 and 2018. Cases lacking a complete address in the SINAN database were excluded from the study.

### 2.4. Data Collection

For data collection, we used the database made available by the Pará State Department of Public Health (“*Secretaria de Saúde Pública do Estado do Pará*”—SESPA). The number of TB cases in Ilha do Marajó was 749; 17% (124) of cases were reported in a municipality other than that of residence. According to the Brazilian National Health System Information Technology Department (DATASUS). The Ilha do Marajó has 16 municipalities, shown in Figure 1.

### 2.5. Data Analysis

The municipality of residence and notification were geocoded with Google Maps and OpenStreetMap (www.openstreetmap.com, accessed on 20 February 2023) using ArcGIS 10.4 software (Esri, Redlands, CA, USA) (https://www.arcgis.com/, accessed on 20 February 2023). We obtained the state, municipal, micro and mesoregion limits from the Brazilian Institute of Geography and Statistics (IBGE) [6].

In the patient flow map, the colours of the lines represent the flow intensity between the municipality, which is the number of patients notified in another municipality. We established four thorough quartiles: light green (small flow: one case), dark green (medium flow: two to four cases), yellow (intermediate flow: five to eight cases), and red (intense flow: nine to fourteen cases).

Notification of immigrant cases and the number of cases of residents notified by health establishments (residents of the municipality) were geocoded, divided into quartiles: 1 to 7 cases; 8 to 40; 41 to 85 cases; above 85 cases.

We extracted the information on health establishments from the National Register of Health Establishments (CNES) (www.cnes.datasus.gov.br, accessed on 20 February 2023), and only the notifying units were geocoded by Google Maps using ArcGIS 10.4 software. We considered primary care establishments (outpatient clinics, health posts, health centres, and mobile river units) as well as medium/high care (hospitals and mixed units). We used Kernel’s estimate for patients who sought other healthcare facilities. Thus, it was possible to analyse the behaviour of the point patterns and provide the point intensity of the process throughout the studied area through Interpolation. It was applied using the parameters of the quartic function and 1.5 km radius, divided between absent (dark green), low (light green), medium (yellow), high (orange), and very high (red).

We obtained the global association between variables through principal component analysis (PCA). For the tabulation of data, we used the statistical programme Statistical Package for the Social Science (SPSS) and EpiInfo v7.2 in a Windows 7 environment. For the descriptive analysis, we performed Pearson’s Chi-square test (Wilks’ G2) for independence, adopting a significance level of *p*-value < 0.05 and used Test G for variables with *n* < 5 and *p*-value < 0.05. Correlation between the variables of the same sample and comparison between the two groups of samples was performed.

### 2.6. Ethics

The study was approved by the Research Ethics Committee of the Universidade do Estado Pará (UEPA) under the number 3,705,199.

## 3. Results

From SINAN, we collected 124 cases which referred to locations beyond the patient’s municipality of residence. Among these cases, 73% (90/124) were reported in the metropolitan region of Belém, which includes Belém, Marituba, and Ananindeua. Additionally, 2% (3/124) were reported in the northeast region of Pará, covering Abaetetuba, Barcarena, and Limoeiro do Ajuru. The remaining 25% (31/124) were reported in various municipalities within Marajó.

Table 1 illustrates the distribution of notifications across various municipalities based on the patient’s city of residence. In Afuá, all new TB cases (n = 7) were reported within the municipality. In Anajás, 90% (n = 36) of new patients were reported within the municipality, while the remaining 10% were notified in Belém (5%; n = 2) and Breves (5%; n = 2). The weight of each column in the table represents the proportion of new cases in the entire dataset. Notably, the proportion of new cases in Breves is 0.234%, indicating that this municipality accounts for 23.4% (n = 165) of all new TB cases in the dataset.

Table 2 presents the sociodemographic characteristics of all patients notified in a municipality where they are not residents. The highest number of cases was reported in 2014 (25; 20%); throughout the 6 years, the average number of cases per year remained close to 20.66 (σ = 3.8297). Most of the 124 reported cases were male (76; 61%). The age group between 20 and 39 years old accounted for 48 cases (39%), with an average age of 37 years (σ = 19.0087) ranging from 1 to 79 years. In terms of education, 46 cases (37%) had incomplete elementary education, 2 cases (2%) had completed higher education, and 13 cases (10%) were illiterate. Most patients (106; 85%) identified themselves as “brown” regarding ethnicity. All variables showed statistical significance (*p*-value < 0.05). Upon comparing the two sample groups, it is evident that there is no statistical difference in sociodemographic variables.

Regarding the clinical characteristics, 78% (97/124) were pulmonary TB (Table 3). Among the cases of extrapulmonary and mixed infection, 30% (8/27) were pleural, 22% (6/27) military, 22% (6/27) ganglionic, 7% (2/27) meningoencephalitis, 4% (1/27) bone, 11% (3/27) characterised as “other” (two disseminated and one mesenteric), and 4% (1/27) case of infection site not found.

Despite the notified cases, 25 (20%) patients did not undergo diagnostic bacterioscopy, and 30 (24%) reported a negative result. Of the patients with negative bacterioscopy (30), only twelve (40%) underwent culture, four (13%) were positive, two (7%) were negative, and six (20%) were being analysed at the time. Of the 124 cases, 23 (18%) underwent culture, 6 (5%) underwent RMT for TB, 2 (2%) of which were susceptible to rifampicin, and 2 (2%) were inconclusive.

When examining the association with directly observed treatment (DOT), 16 cases (31%) were recorded as following this treatment approach, while 51 cases (41%) did not adhere to this modality, and 57 cases (46%) were not even registered in the notification form. Regarding the outcome, more than half of the cases achieved cure (55%). In comparison, there were 9 cases (7%) of treatment dropout, 4 cases (3%) resulting in death due to TB, 1 case (1%) of drug-resistant TB, and 36 cases (29%) were transferred to their respective municipality’s health unit for continued treatment after diagnosis.

Figure 2 illustrates the patient flow of individuals residing on Ilha do Marajó, who were notified in other municipalities both within the island and the state. In this network representation, each node corresponds to a municipality where the patients live, and the connecting lines depict the flow of patients, quantitatively weighted based on those received in another municipality during the specified period.

The most significant patient flow is directed towards the state capital, Belém, primarily originating from the municipalities of Soure (n = 14; 18%), Salvaterra (n = 12; 24%), Ponta de Pedras (10 cases; 25%), and São Sebastião da Boa Vista (n = 10; 20%). These municipalities are in closer geographical proximity to the capital. Another noticeable flow is from Bagre to Breves, where nine cases (58.3%) of Bagre residents were reported in Breves. Additionally, there are intermediate flows, such as from Santa Cruz do Arari to Belém (n = 6; 22%), Portel to Belém (n = 6; 9.3%), and Melgaço to Breves (n = 5; 19%).

Ilha do Marajó comprises 238 health facilities, out of which 67 facilities (28%) were notified as active, and 3 were inactive, specifically located in the municipality of Breves. However, even during their inactive status, there were cases reported for the period they were active. These health establishments were categorised into two groups: Basic Care facilities, including ambulatory, health centres, health posts, and mobile river units, which accounted for 223 units (93%), and Medium/High complexity facilities, including mixed units and hospitals, which represented 18 establishments (7%). Among these establishments, 8 operate under dual management (state and municipal), 1 is under state management, and the majority, 232 facilities, are solely under municipal control (Figure 2).

The municipality with the highest number of health facilities was Breves, with 36 active and 3 inactive units, but only 5 units reported cases. Within the island, this municipality receives the most patients from other municipalities. The municipality with the lowest number was Santa Cruz do Arari, having six establishments (one notified), with 22% of its residents reported in Belém.

Of the municipalities outside the Ilha do Marajó, the capital Belém had 19 establishments notifying residents of Marajó, with 11 (58%) of primary care and eight (42%) of medium/high complexity, making it the hospital of reference for infectious and contagious diseases that reported the most cases (37; 44%) and the second was a unit specialised in particular infectious and parasitic diseases (18; 21%).

## 4. Discussion

Exploring social networks in the health sciences field originated from comprehending the dynamics of microorganism transmission within confined spaces, specifically through the interactions between their components (patients) and their movements in the area. According to Nilo et al. [9], this analytical approach in public health facilitates the identification of individuals and/or clusters that are particularly susceptible to transmission, as they possess a higher number of connections as vulnerable nodes. This identification serves as an alert for the requirement of targeted local interventions to control and prevent the disease.

The clinical and epidemiological characteristics observed in this study resemble those identified in a previous study by Mesquita et al. [10] on TB cases in Ilha do Marajó. These characteristics include a predominance of male patients, individuals in the adult age group, low education levels, and the pulmonary clinical form. Similar patterns have also been reported in other studies conducted in Brazil [5,11]. However, these sociodemographic variables did not demonstrate statistical significance regarding patient mobility for seeking care. In other words, the movement between municipalities was independent of these variables, except for the ethnicity variable. Notably, individuals who identified as “brown” exhibited the highest level of mobility between municipalities.

TB continues to possess stigmatising characteristics, shedding light on the precarious social conditions, including living conditions [12], experienced by many patients. Therefore, conducting a social network analysis is crucial, as understanding the population’s inclination to seek improved healthcare directly impacts the dynamic circulation of *Mycobacterium tuberculosis*. One of the critical foundations of the PNCT is the decentralisation of TB control actions to PHC units. These units are deemed the focal point for managing all prevention and control measures, ensuring comprehensive and continuous care. These measures include actively searching for suspected cases displaying symptoms, conducting diagnostic investigations, monitoring confirmed cases, supervising treatment, and performing regular bacterioscopy for control [12].

In this study, only 16 patients (13%) entered the DOT mode, and in the state of Pará in 2022, 25.1% of patients in were DOT [13]. This fact reinforces the notion that more decentralised PHC action may be lacking to strengthen treatment adherence, prevent drug-resistant TB cases, reduce treatment dropout rates, and increase cure [14]

Following the guidelines set by the World Health Organization (WHO) for tuberculosis (TB) control, it is recommended to attain a cure rate of 85% or higher while keeping the abandonment rate below 5% [15]. Regrettably, this study did not observe the fulfilment of the proposed cure goal nor the abandonment goal of 7% (with a sub-optimal rate of only 45%). These outcomes possibly indicate inadequate epidemiological surveillance in managing TB cases, as the observed rates fell significantly below the Brazilian Ministry of Health recommendations. It is plausible that limitations in patient accessibility have contributed to these deviations.

Ilha do Marajó is renowned for its extractive production (fruit and fisheries) and livestock farming. However, it is crucial to acknowledge that this region is characterised by extreme poverty, with municipalities exhibiting the lowest HDI in Brazil [16,17]. When a municipality lacks adequate PHC coverage, residents often seek assistance from neighbouring municipalities. According to the Brazilian Ministry of Health guidelines, the first sputum collection should occur when the patient visits the health unit for service. Furthermore, patients are instructed to perform the second collection at home in the early morning. This approach ensures that individuals residing in rural areas or farther away from healthcare units can undergo the necessary examinations, even if they cannot return the following day. However, during field trips, it was observed that some healthcare units neglect the initial sputum collection and only offer limited appointment availability, typically restricted to one or two days per week.

The transmission dynamics of tuberculosis (TB) are influenced by the shift within the region due to its aerosol-transmissible nature. This area’s central displacement mode relies on river transport, with rivers and boats serving as the primary means of transportation for daily activities. However, river transport in Pará is often overcrowded, exceeding seating capacity and resulting in extended travel times between municipalities. This crowded and prolonged travel environment significantly facilitates the spread of infectious diseases.

The combination of unstable social policies and limited healthcare services in Ilha do Marajó leads TB patients to seek primary healthcare in neighbouring municipalities, creating spatial heterogeneity. This spatial heterogeneity refers to the transmission of infectious diseases related to travel. Apart from intensifying the transmission chain, the dispersion of infected individuals hampers disease control compared to cases that are not dispersed [3,18,19].

In Ilha do Marajó, we observed patient movement patterns towards the municipality of Breves, which hosts many healthcare facilities, including those offering medium/high complexity services close to the state capital, Belém. Regarding Belém, we noticed a higher concentration of TB cases in specialised sites for infectious diseases, which raises the likelihood of transmission chains during this displacement, mainly due to long sea journeys where individuals are confined on boats for hours until reaching their destination.

According to Azevedo et al. [12], patients tend to believe that seeking care at Emergency Care Units (ECUs) is the best option to promptly address their health concerns, often lacking awareness of PHC facilities. This situation impacts TB diagnosis, as patients seeking care at urgent care institutions (ECUs and/or referral hospitals) receive treatment for their isolated symptoms without resolving the underlying issue. In Ilha do Marajó, given the relatively low population in some municipalities, not everyone has access to urgent and emergency services, which further exacerbates the problem of patient displacement.

It is worth emphasising that a strategy capable of effectively reducing the number of patients seeking emergency care involves conducting home visits and providing guidance to community health agents. Assume these agents become aware of or identify someone experiencing a cough lasting three weeks or longer, in this case, referring that individual to a healthcare facility for necessary examinations is essential. Additionally, fostering a stronger connection between the health service, healthcare professionals, individuals, families, and the community plays a crucial role in this approach [19].

## 5. Conclusions

Understanding the organisation of PHC and the search for care in TB transmission is a central factor in controlling diseases in risky places. This study, conducted in Ilha do Marajó, revealed a noteworthy pattern wherein patients from this region frequently seek TB treatment outside their home municipalities. This trend is attributed to limitations in the healthcare services provided by the municipalities, emphasising the importance of addressing these deficiencies to combat the spread of the disease effectively.

A notable finding of this study is that it did not observe the achievement of the established 85% cure goal, and the dropout rate surpassed the threshold of 5%, indicating inadequate epidemiological surveillance in managing TB cases in this specific region. Crucially, the effective organisation of patient flow within the PHC in each municipality plays a vital role in fostering the dedication of health teams and facilitating coordinated care, ultimately leading to the achievement of TB control objectives. By establishing well-structured service networks, it becomes possible to analyse TB’s social, clinical, and epidemiological aspects, enabling the identification of risk groups and regions with a higher patient flow. This approach is precious in addressing disease outbreaks in vulnerable areas with neglected populations.

## Figures and Tables

**Figure 1 tropicalmed-08-00397-f001:**
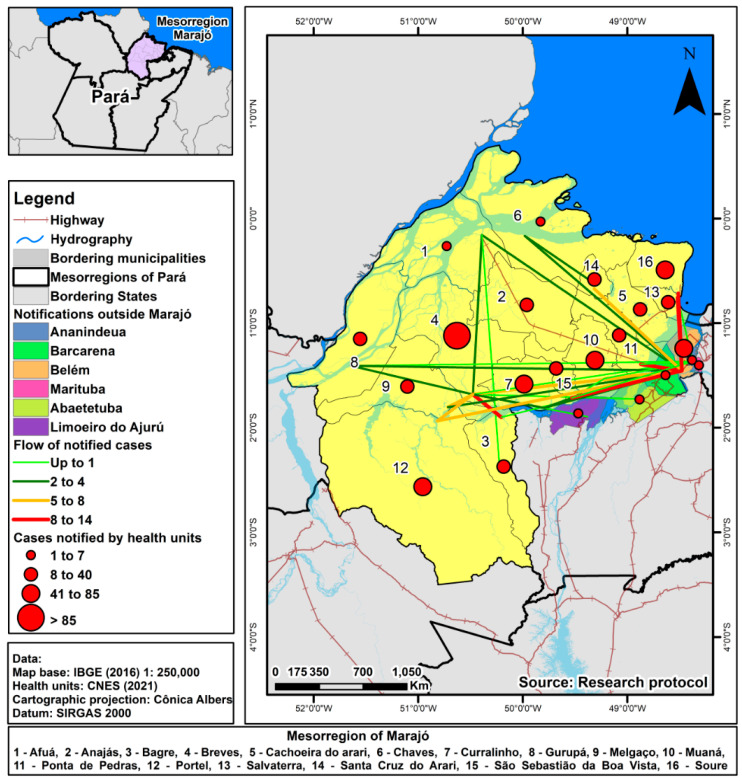
The flow of tuberculosis patients in Ilha do Marajó and neighbouring municipalities, 2013–2018.

**Figure 2 tropicalmed-08-00397-f002:**
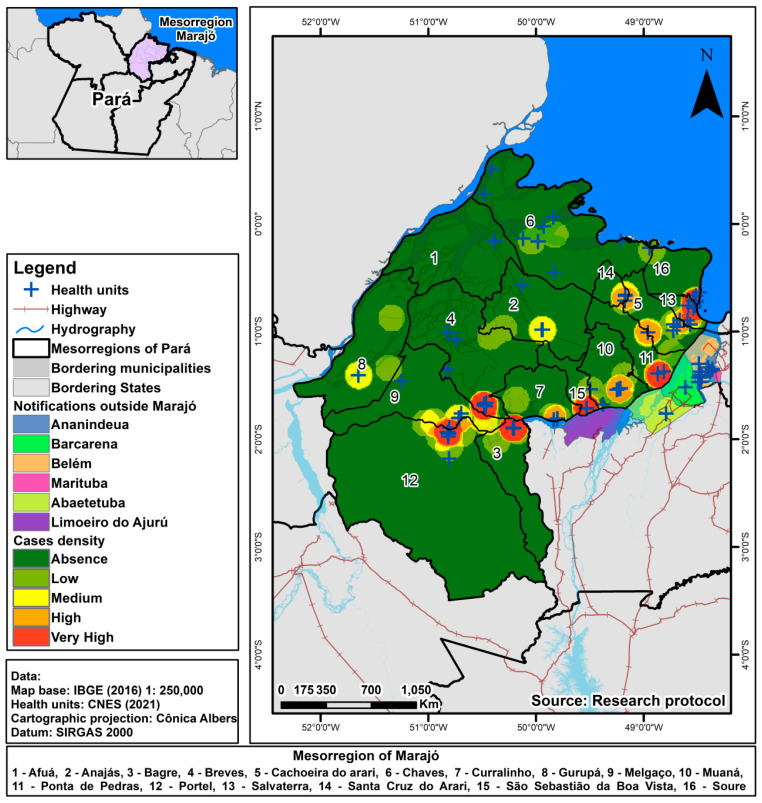
Kernel estimate of TB cases reported in other counties and the health facilities that reported the cases.

**Table 1 tropicalmed-08-00397-t001:** Line profile between the municipality of notification and the municipality of residence of new human tuberculosis cases in Ilha do Marajó, from 2013 to 2018.

Municipality of Residence (New Tuberculosis Cases)																
Municipality Notification	Afuá	Anajás	Bagre	Breves	Cachoeira do Arari	Chaves	Curralinho	Gurupá	Melgaço	Muaná	Ponta de Pedras	Portal	Salvaterra	Santa Cruz. do Arari	São S. da Boa Vista	Source
Abaetetuba	0.00%	0.00%	0.00%	0.60%	0.00%	0.00%	0.00%	0.00%	0.00%	0.00%	0.00%	0.00%	0.00%	0.00%	0.00%	0.00%
Ananindeua	0.00%	0.00%	0.00%	0.60%	0.00%	0.00%	0.00%	0.00%	0.00%	0.00%	0.00%	0.00%	0.00%	0.00%	0.00%	0.00%
Afuá	100.00%	0.00%	0.00%	0.00%	0.00%	0.00%	0.00%	0.00%	0.00%	0.00%	0.00%	0.00%	0.00%	0.00%	0.00%	0.00%
Anajás	0.00%	90.0%	0.00%	0.00%	0.00%	0.00%	0.00%	0.00%	0.00%	0.00%	0.00%	0.00%	0.00%	0.00%	0.00%	0.00%
Barcarena	0.00%	0.00%	4.20%	0.00%	0.00%	0.00%	0.00%	0.00%	0.00%	0.00%	0.00%	0.00%	0.00%	0.00%	0.00%	0.00%
Bagre	0.00%	0.00%	58.30%	0.00%	0.00%	0.00%	0.00%	0.00%	0.00%	0.00%	0.00%	0.00%	0.00%	0.00%	0.00%	0.00%
Belém	0.00%	5.00%	0.00%	2.90%	20.00%	28.60%	5.40%	8.80%	7.70%	10.30%	25.00%	9.40%	24.00%	22.20%	19.60%	15.80%
Breves	0.00%	5.00%	37.50%	95.40%	0.00%	0.00%	2.70%	8.80%	19.20%	0.00%	0.00%	9.40%	0.00%	0.00%	0.00%	0.00%
Cachoeira do Arari	0.00%	0.00%	0.00%	0.00%	80.00%	0.00%	0.00%	0.00%	0.00%	0.00%	0.00%	0.00%	0.00%	0.00%	0.00%	0.00%
Chaves	0.00%	0.00%	0.00%	0.00%	0.00%	42.90%	0.00%	0.00%	0.00%	0.00%	0.00%	0.00%	0.00%	0.00%	0.00%	0.00%
Curralinho	0.00%	0.00%	0.00%	0.00%	0.00%	0.00%	86.50%	0.00%	0.00%	0.00%	0.00%	0.00%	0.00%	0.00%	0.00%	0.00%
Gurupá	0.00%	0.00%	0.00%	0.00%	0.00%	0.00%	0.00%	79.40%	0.00%	0.00%	0.00%	0.00%	0.00%	0.00%	0.00%	0.00%
Limoeiro do Ajuru	0.00%	0.00%	0.00%	0.60%	0.00%	0.00%	0.00%	0.00%	0.00%	0.00%	0.00%	0.00%	0.00%	0.00%	0.00%	0.00%
Marituba	0.00%	0.00%	0.00%	0.00%	0.00%	0.00%	0.00%	0.00%	0.00%	0.00%	2.50%	0.00%	0.00%	0.00%	0.00%	0.00%
Melgaço	0.00%	0.00%	0.00%	0.00%	0.00%	0.00%	0.00%	0.00%	73.10%	0.00%	0.00%	0.00%	0.00%	0.00%	0.00%	0.00%
Muaná	0.00%	0.00%	0.00%	0.00%	0.00%	0.00%	0.00%	0.00%	0.00%	86.20%	0.00%	0.00%	0.00%	0.00%	0.00%	0.00%
Ponta de Pedras	0.00%	0.00%	0.00%	0.00%	0.00%	0.00%	0.00%	0.00%	0.00%	0.00%	72.50%	0.00%	0.00%	0.00%	0.00%	0.00%
Portel	0.00%	0.00%	0.00%	0.00%	0.00%	0.00%	0.00%	0.00%	0.00%	0.00%	0.00%	81.30%	0.00%	0.00%	0.00%	0.00%
Salvaterra	0.00%	0.00%	0.00%	0.00%	0.00%	0.00%	0.00%	0.00%	0.00%	0.00%	0.00%	0.00%	74.00%	0.00%	0.00%	0.00%
Santa Cruz do Arari	0.00%	0.00%	0.00%	0.00%	0.00%	28.60%	0.00%	0.00%	0.00%	0.00%	0.00%	0.00%	0.00%	77.80%	0.00%	0.00%
São Sebastiao da Boa Vista	0.00%	0.00%	0.00%	0.00%	0.00%	0.00%	5.40%	0.00%	0.00%	1.70%	0.00%	0.00%	0.00%	0.00%	78.40%	0.00%
Soure	0.00%	0.00%	0.00%	0.00%	0.00%	0.00%	0.00%	0.00%	0.00%	0.00%	0.00%	0.00%	0.00%	0.00%	0.00%	84.20%
Weight	0.90%	5.30%	3.20%	23.40%	4.70%	0.90%	4.90%	4.50%	3.50%	7.70%	5.30%	8.50%	6.70%	3.60%	6.80%	10.10%

**Table 2 tropicalmed-08-00397-t002:** The sociodemographic profile of all new human tuberculosis cases in Ilha do Marajó and those notified outside their municipality of residence, 2013–2018.

Diagnostic Year	n (749)	%	n (124)	%	*p*-Value ^a^
2013	103	14	16	13	
2014	120	16	25	20	
2015	120	16	17	14	
2016	127	17	23	19	0.37 ^1^
2017	141	19	24	19	
2018	138	18	19	15	
**Sex**	**n (749)**	**%**	**n (124)**	**%**	
Male	487	65	76	61	0.48 ^1^
Female	262	35	48	39	
**Age range**	**n (749)**	**%**	**n (124)**	**%**	
<1	1	1	0	0	
1 to 19	106	14	21	17	
20 to 39	334	45	48	39	0.78 ^2^
40 to 59	190	25	34	27	
>60	110	15	21	17	
**Education**	**n (749)**	**%**	**n (124)**	**%**	
Illiterate	76	10	13	10	
Incomplete Elementary School	361	48	46	37	0.42 ^2^*
Complete Elementary School (8th grade)	35	5	8	6	
Incomplete High School	59	8	8	6	
Complete High School	53	7	10	8	
Incomplete Higher Education	7	1	2	2	
Complete Higher Education	12	2	2	2	
Ignored	146	19	35	29	
**Ethnicity**	**n (749)**	**%**	**n (124)**	**%**	
White	65	9	3	2	
Black	57	8	7	6	
Brown (“Parda”)	611	82	106	85	0.00 ^2^*
Ignored	10	1	8	7	

^a^ correlation between two populations. ^1^ Pearson’s chi-square test (Wilks’ G^2^) for independence (*p*-value < 0.05). ^2^ Test G (*p*-value < 0.05). * Significant Values.

**Table 3 tropicalmed-08-00397-t003:** Clinical profile of all new human tuberculosis cases in Ilha do Marajó and notified cases outside the municipality of residence, 2013–2018.

Variables	n (749)	%	n (124)	%	*p*-Value ^1^
**Clinic characteristics**					
Pulmonary	686	92	97	78	
Extrapulmonary	43	5.8	16	13	<0.00 ^2^*
Pulmonary and Extrapulmonary	19	2	11	9%	
**Diagnostic by bacterioscopy**					
Positive	531	71	69	56	
Negative	121	16	30	24	0.00 ^1^*
Not performed	97	13	25	20	
**Culture**					
Positive	64	0	9	7	
Negative	17	2	3	2	0.00 ^2^*
Ongoing	15	2	11	9	
Not performed	652	87	101	82	
**Histopathologic**					
Positive bacterioscopy	137	18	3	2	
Suggestive of tuberculosis	21	3	3	2	
Not suggestive of tuberculosis	11	1.5	3	2	<0.00 ^2^*
Ongoing	43	5.8	5	4	
Not performed	498	66.5	110	90	
Unknown	39	5.2	0	-	
**Tuberculosis Rapid Molecular Test**					
Detectable—rifampicin susceptible	8	1	4	3	
Detectable—rifampicin resistant	2	0.2	0	0	<0.00 ^2^*
Not performed	465	62.1	79	63	
Not Detectable	8	1	0	-	
Inconclusive	16	2.2	2	2	
Unknown	250	33.5	40	32	
**Directly Observed Therapy**					
Yes	164	22	16	13	
No	383	51	51	41	<0.00 ^1^*
Ignored	202	27	57	46	
**Outcome**					
Cure	499	67	55	45	
Abandonment	88	12	9	7	
Death from tuberculosis	23	3	4	3	<0.00 ^2^*
Death from other causes	27	2	8	6	
Transfer	64	9	36	29	
Drug-resistant tuberculosis	8	1	1	1	
Treatment scheme change	3	0.4	1	1	
Treatment failure	2	0.3	0	0	
Change of diagnosis	3	0.4	0	0	
Unknown	32	4	10	8%	

^1^ Pearson’s Chi-square test (Wilks’ G^2^) for independence (*p*-value < 0.05). ^2^ Test G (*p*-value < 0.05). * Significant Values.

## Data Availability

All relevant data is presented within the manuscript.

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
