# Peer review of "Spatial Analysis of Tuberculosis Patient Flow in a Neglected Region of Northern Brazil"

_tropicalmed, 2023, doi:10.3390/tropicalmed8080397_

Round 1
Reviewer 1 Report
Dear authors,
The article is interesting, but the english must be improved.
Also, I would like to suggest some minor changes to improve the article.
I would suggest separating the methodology into several topics such as: place of study, data collection, data analysis, ethical aspects, etc., for better scientific soundness.
the last paragraph of conclusion: "An important fact, this study did not reach the established goal of cure (85%) and abandonment rate is not less than 5%, which may indicate a deficient epidemiological surveillance in the control of TB cases, because it was much lower than that recommended by the Brazillian Ministry of Health, which may be influenced by limitations on patient care in local health services." would suit better at discussion.
A problem is stated in discussion: "many patients seek emergency care units.." what could be done to reduce this demand and improve the dynamics of tuberculosis at the primary level? I think this information should be discussed.
Best regards,
Moderate english review.
Author Response
RESPONSE TO REVIEWER 1
Point 1: I would suggest separating the methodology into several topics such as: place of study, data collection, data analysis, ethical aspects, etc., for better scientific soundness.
Response 1: Suggestion accepted.
Point 2: The last paragraph of conclusion: "An important fact, this study did not reach the established goal of cure (85%) and abandonment rate is not less than 5%, which may indicate a deficient epidemiological surveillance in the control of TB cases, because it was much lower than that recommended by the Brazillian Ministry of Health, which may be influenced by limitations on patient care in local health services." would suit better at discussion.
Response 2: Suggestion accepted. We moved the results interpretation to the discussion and we kept only the principal conclusion.
Point 3: A problem is stated in discussion: "many patients seek emergency care units.." what could be done to reduce this demand and improve the dynamics of tuberculosis at the primary level? I think this information should be discussed.
Response 3: Suggestion accepted. We added an additional last paragraph within the discussion section.
Reviewer 2 Report
Dear authors,
I have carefully read your manuscript. Data presented is of great value for the healthcare system in Brazil and in other countries. However, some issues must be considered.
The manuscript structure needs to be improved. The aim of the research is not well defined so that, the results do not answer a question. Moreover, the introduction do not provide the enough support to the discussion. The article must focus to some issues and, it appears to simply comment on lots of information found during data analysis.
Also, English must to be improved. The article is very difficult to understand.
For these reason, I recommend the authors to perform a major review starting by looking for a specific aim to be answered and submitting the article to a professional English review or to a native speaker review.
I reinforce that the data found is interesting and important for operational decisions to end TB.
Yours sincerely,
English is very difficult to understanding. Professional English editing or, at least, a native speaker review is necessary.
Author Response
RESPONSE TO REVIEWER 2
Point 1: The aim of the research is not well defined so that, the results do not answer a question
Response 1: Suggestion accepted. We improved the study objective description (last paragraph of the introduction section).
Point 2: Moreover, the introduction do not provide the enough support to the discussion. The article must focus to some issues and, it appears to simply comment on lots of information found during data analysis.
Response 2: To address this suggestion and improve the introduction, a paragraph was added (3rd paragraph). Additionally, the 4th paragraph highlight information about the flow of care for individuals with TB (the city where they live to the city where they were treated). This flow of demand for care can lead to a greater spread of the disease.
Reviewer 3 Report
The work is very interesting and well-prepared. My only objection is the quality of the translation of Part 3 (results), in the section regarding the text lines 141-162. Some sentences are not understandable. There are also terms that are not used in papers on TB, e.g. in the text line 148, instead of 'ganglionic', "lymph node T"B is used; in line 151, the sentence " Despite cases being notified and confirmed" should be changed, the word "confirmed" would be better to take away, because these cases were not confirmed according to accepted definitions. The term "bacilloscopy" is usually replaced by the word "bacterioscopy" in the literature.
I am not qualified to assess the quality of English in this paper but as I wrote above, the "Results" section should be linguistically improved
Author Response
RESPONSE TO REVIEWER 3
Point 1: My only objection is the quality of the translation of Part 3 (results), in the section regarding the text lines 141-162. Some sentences are not understandable. There are also terms that are not used in papers on TB, e.g. in the text line 148, instead of 'ganglionic', "lymph node T"B is used; in line 151, the sentence " Despite cases being notified and confirmed" should be changed, the word "confirmed" would be better to take away, because these cases were not confirmed according to accepted definitions. The term "bacilloscopy" is usually replaced by the word "bacterioscopy" in the literature.
Response 1: Suggestion accepted and all corrected. All changes made to the article are highlighted in green so that reviewers and editors can identify. Finally, extensive correction of the English language was carried out.
